# Evaluation of Anticancer and Anti-Inflammatory Activities of Some Synthetic Rearranged Abietanes

**DOI:** 10.3390/ijms241713583

**Published:** 2023-09-01

**Authors:** Mustapha Ait El Had, Houda Zentar, Blanca Ruiz-Muñoz, Juan Sainz, Juan J. Guardia, Antonio Fernández, José Justicia, Enrique Alvarez-Manzaneda, Fernando J. Reyes-Zurita, Rachid Chahboun

**Affiliations:** 1Departamento de Química Orgánica, Facultad de Ciencias, Instituto de Biotecnología, Universidad de Granada, 18071 Granada, Spain; mustapha.aitelhad20@gmail.com (M.A.E.H.); zentarhouda@correo.ugr.es (H.Z.); pepojaen@hotmail.com (J.J.G.); ajfvargas@ugr.es (A.F.); jjusti@ugr.es (J.J.); eamr@ugr.es (E.A.-M.); 2Departamento de Bioquímica y Biología Molecular I, Facultad de Ciencias, Universidad de Granada, 18071 Granada, Spain; blancaruiz_@uma.es (B.R.-M.); jsainz@ugr.es (J.S.); 3Centre for Genomics and Oncological Research: Pfizer, Genomic Oncology Area, GENYO, University of Granada, Andalusian Regional Government, PTS Granada, 18016 Granada, Spain

**Keywords:** rearranged abietane type, semisynthesis, antitumor activity, apoptosis, colorectal cancer, anti-inflammatory activity, nitric oxide

## Abstract

Synthesis of the rearranged abietane diterpenes pygmaeocins C and D, viridoquinone, saprorthoquinone, and 1-deoxyviroxocine has been successfully achieved. The anticancer and anti-inflammatory activities of selected orthoquinonic compounds **5**, **7**, **13**, and **19**, as well as pygmaeocin C (**17**), were evaluated for the first time. The antitumor properties were assessed using three cancer cell lines: HT29 colon cancer cells, Hep G2 hepatocellular carcinoma cells, and B16-F10 murine melanoma cells. Compounds **5** and **13** showed the highest cytotoxicity in HT29 cells (IC_50_ = 6.69 ± 1.2 µg/mL and IC_50_ = 2.7 ± 0.8 µg/mL, respectively). Cytometric studies showed that this growth inhibition involved phase S cell cycle arrest and apoptosis induction, possibly through the activation of the intrinsic apoptotic pathway. Morphological apoptotic changes, including nuclear fragmentation and chromatin condensation, were also observed. Furthermore, the anti-inflammatory activity of these compounds was evaluated on the basis of their ability to inhibit nitric oxide production on the lipopolysaccharide activated RAW 264.7 macrophage cell line. Although all compounds showed high anti-inflammatory activity, with percentages between 40 and 100%, the highest anti-inflammatory potential was obtained by pygmaeocin B (**5**) (IC_50NO_ = 33.0 ± 0.8 ng/mL). Our results suggest that due to their dual roles, this type of compound could represent a new strategy, contributing to the development of novel anticancer agents.

## 1. Introduction

Cancer is currently one of the most common causes of death worldwide, with its prevalence projected to reach 21 million cases by 2030 [1]. The current chemotherapeutic drugs used in cancer treatment often have severe side effects and can lead to resistance development. Thus, there is an urgent need for novel and potent antitumor agents with reduced toxicity. Natural products, which have been used for centuries to prevent and treat human diseases, offer a promising avenue for the discovery of such active compounds. The plant kingdom serves as an abundant source of bioactive compounds with exceptional therapeutic potential [2], and over 25% of approved drugs in cancer therapy are derived from natural sources [3]. Plants, the basis of traditional medicine, are instrumental in the search for chemical structures that have led to the discovery of new drugs. These compounds, known as secondary metabolites, are synthesized by plants in response to various environmental stresses, enabling their interaction with their surroundings and contributing to their survival [4]. Notably, among these metabolites are abietanes, a vast group of diterpenes displaying a wide range of biological activities [5]. Abietanes with a phenanthrene skeleton are the most common group of this class of diterpenes. Thus, ferruginol (**1**) [6], which contains a tricyclic structure with an aromatic C-ring, is a classic example (Figure 1). Seco-abietanes with an aromatic B-ring are derivatives resulting from the loss of a methyl group on C-10. A prominent example of this peculiar group of *nor*-diterpenes is represented by tanshinone IIA (**2**), which has been shown to exert a proapoptotic function against the MCF–7 breast cancer cell line by inhibiting the downstream signaling of protein kinase C [7].

On the other hand, a crucial aspect of this research is the relationship between chronic inflammation and tumor development [8], which can lead to various diseases, including cancer. As a consequence, there is a strong interest in developing novel anti-inflammatory agents with potent anticancer properties [9,10]. In line with this information, our ongoing efforts to discover new potential drugs have led us to synthesize natural abietane and rearranged abietanes and evaluate their anticancer and anti-inflammatory activities.

Rearranged abietane-type diterpenes constitute a reduced group of secondary metabolites that are structurally and biologically very interesting. According to the structures of their skeletons, they can be divided into five subgroups: Taiwaniquinone D (**3**) [11] (type 1) and barbatusol (**4**) [12] (type 2) are two examples of the rearranged abietane family. The former has a contracted B-ring of the abietane skeleton, whereas the latter has an expanded B-ring with a 9(10-20) abeo-abietane skeleton called icetexane. The other three types are exemplified by Pygmaeocin B (**5**) [13] (type 3), which is characterized by a methyl C-20 transposed from C-10 to C-5; umbrosone (**6**) [14], with a phenanthrene skeleton; and saprorthoquinone (**7**) (type 5), a 4,5-seco-abietane with an *o*-naphthoquinone structure isolated from the roots of *Salvia prionitis* [15] and exhibiting strong cytotoxicity against P388 leukemia cells [16].

Pygmaeocin B (**5**) [13] (type 3), with methyl C-20 transposed from C-10 to C-5, and umbrosone (**6**) [14] (type 4), with a phenanthrene skeleton, are two other rearranged abietanes (Figure 1). Finally, saprorthoquinone (**7**) (type 5), a bicyclic naphthoquinone with a rearranged abietane structure abietane diterpenoid, which was isolated from the roots of *Salvia prionitis* [15], exhibits strong cytotoxicity against P388 leukemia cells [16]. This last type class of rearranged abietane can undergo biological transformation to other related tricyclic metabolites, such as ceratodiol (**8**) [17,18], saprirearine (**9**) [18], microstegiol (**10**) [18], candidissiol (**11**) [18], and 1-deoxyviroxocin (**12**) [17,19,20] (Figure 2). 1-Deoxyviroxocin (**12**), with an oxocane ring, is an unusual oxygenated eight-membered ring-type oxocane and was isolated from different natural sources such as the cones of *Taxodium distichum* L. [17], a conifer widely distributed in the southeastern United States. Furthermore, 1-deoxyviroxocin (**12**) has also been isolated from *Caryopteris incana* (Thunb.) [20] and from the roots of *Zhumeria majdae*, an endemic Iranian plant belonging to the Lamiaceae family with anti-inflammatory, antiprotozoal, and anticonvulsant properties [19].

This study presents the synthesis of the natural compounds pygmaeocins B (**5**) and C (**17**), viridiquinone (**19**), saprorthoquinone (**7**), and 1-deoxyviroxocin (**12**) from ferruginol (**1**). Moreover, we conducted a biological evaluation of the natural and synthetic compounds, which include pygmaeocins B (**5**) and C (**17**), viridoquinone (**19**), saprortoquinone (**7**), and the syntetical *ortho*quinone **13**. These selected compounds were evaluated for their inhibitory activity against nitric oxide (NO) production in the lipopolysaccharide (LPS)-induced murine macrophage RAW 264.7 cell line, as well as for their antiproliferative activity against colon cancer cells (HT29), melanoma cells (B16–F10), and hepatoma cells (HepG2). Furthermore, cytometric and microscopic assays were performed on the most active compounds pygmaeocin B (**5**) and its precursor **13**.

## 2. Results

### 2.1. Chemistry

During our studies on the synthesis of abietanes, a new synthetic methodology was developed and applied to the preparation of some rearranged abietanes type 2, such as pygmaeocin B (**5**), pygmaeocin C (**17**), and viridoquinone (**19**) [21] from ferruginol (**1**). The first step involved the oxidation of **1** to orthoquinone **13** with (PhSeO)_2_O, which after treatment with Sc(OTf)_3_ in the presence of Ac_2_O generated the derivative **14**. This compound was transformed into **16** by the action of SeO_2_ under reflux in dioxane and subsequent oxidation with PDC. Deprotection of catechol **16** with conc. HCl generated pygmaeocin C (**17**), which was easily oxidized to pygmaeocin B (**5**) using Ag_2_O (see Figure 1).

Next, viridiquinone (**19**) was successfully obtained in two steps from orthoquinone **13** by isomerization with conc. H_2_SO_4_ and subsequent oxidation with SeO_2_ at reflux in dioxane (see Figure 2).

Treatment of catechol **18** with one equivalent of SeO_2_ in the presence of a catalytic amount of H_2_SO_4_ surprisingly led to saprorthoquinone (**7**), a rearranged abietane-type 5 previously synthesized in eight steps from ferruginol methyl ether, with an overall yield of 24% [22]. When this reaction was carried out with two equivalents of SeO_2_ and 1-eq of H_2_SO_4_, aldehyde **20** was obtained after 15 min of refluxing in dioxane. The treatment of orthoquinone **13** with SeO_2_ and conc. H_2_SO_4_ under stoichiometric amounts leads directly to saprorthoquinone (**7**) after 20 min with a good yield. Thus, the synthesis has been completed in only two steps starting from ferruginol (**1**), with an overall yield of 60% (see Figure 3).

Finally, the treatment of catechol **18** with two equivalents of SeO_2_ at reflux in dioxane in the absence of acid yielded hydroxyenone **21**, which was completely transformed into 1-deoxyviroxocin (**12**) after reaction with I_2_/PPh_3_ for 10 min at room temperature (Figure 3). Therefore, the first synthesis of 1-deoxyviroxocin (**12**) from ferruginol (**1**) was carried out in only four steps. The NMR spectroscopic data of **12** were compared with those described in the literature (see SI). In general, all the obtained signals agreed with the published data, and there were hardly any significant deviations; therefore, the structure of 1-deoxyviroxocin (**12**) was confirmed.

### 2.2. Anticancer Activity

#### 2.2.1. Cell Viability Assay

In this study, we employed the MTT (3-(4,5-dimethyl thiazol-2-yl)-2,5-diphenyltetrazolium bromide) colorimetric assay to evaluate the cytotoxic activity of selected abietanes (compounds **5**, **7**, **13**, **17**, and **19**) against three tumor cell lines: human colon adenocarcinoma (HT29), human hepatocarcinoma (HepG2), and murine melanoma (B16–F10). The MTT assay relied on the conversion of MTT to formazan by viable cells, with the formazan concentration proportional to the number of viable cells. The concentrations required for 50 and 80% growth inhibition (IC_50_ and IC_80_) were determined for each compound.

After 72 h of treatment, all tested compounds demonstrated a dose-dependent decrease in cell viability. Among the abietanes, compound **13**, which serves as the precursor for the synthesis of the selected rearranged abietanes, exhibited the highest cytotoxicity in all three cell lines, with IC_50_ values ranging from 2.67 to 5.58 µg/mL and IC_80_ values in the range of 5.19 to 7.67 µg/mL. Furthermore, pygmaeocin B (**5**) demonstrated substantial cytotoxic activity against the HT29 and HepG2 cell lines, with IC_50_ concentrations of 6.69 and 8.98 µg/mL, respectively. In contrast, Viridoquinone (**19**) displayed much lower toxicity in these two cell lines, with IC_50_ values up to 23 and 14 times higher than that of precursor compound **13**, respectively. The remaining diterpenoids (**5**, **7**, and **17**) exhibited comparable effects on cell growth in the HT29 and HepG2 cell lines (Table 1).

Interestingly, the cytotoxic response of B16–F10 melanoma cell line to the compounds differed significantly. In this case, Viridoquinone (**19**) and Saprorthoquinone (**7**) exhibited high cytotoxicity, with IC_50_ values of 6.42 and 6.89 µg/mL, respectively. However, Pygmaeocin B (**5**) did not significantly reduce the growth of B16–F10 cells, with an IC_50_ value 3.5 times higher than that of compound **13**. Based on their promising cytotoxic activity, pygmaeocin B (**5**) and synthetic orthoquinone **13** were selected for further investigation of the mechanisms involved in HT29 colon adenocarcinoma cell growth inhibition. The dose–response curves for these compounds are illustrated in Figure 3.

These findings reveal the potential of compounds **5** and **13** as plausible candidates for future development in cancer therapy and underscore the importance of further research to elucidate their mechanisms of action in tumor cells. These products were selected for the percentage of apoptosis determination, cell cycle distribution, and mitochondrial membrane potential measurements. All these assays were realized in the HT29 colon adenocarcinoma cell line by flow cytometry analysis in a fluorescence-activated cell sorter.

#### 2.2.2. Cell Cycle Arrest and Distribution

Flow cytometry was employed to assess the impact of compounds **5** and **13** on the cell cycle profiles of HT29 cells after 72 h of treatment; by using propidium iodide (PI) incorporation, whose fluorescence directly correlated with DNA content, we analyzed the distribution of cells in different cell cycle phases. The DNA histogram analysis (Figure 4) demonstrated that both compounds induced cell cycle arrest in the S phase.

Orthoquinone **13** caused a significant increase in the cell population in the S phase, up by 10% compared with the control (untreated cells), regardless of the administered dose. This increase was accompanied by a decrease in the cell populations in the G0/G1 and G2/M phases. On the other hand, Pygmaeocin B (**5**) exhibited a more pronounced arrest of the cell cycle in the S phase at IC_80_ doses compared with IC_50_ doses, leading to increases of 4% and 10%, respectively, compared with the control. However, the changes in the cell number in the G2/M phase were not significant.

These results provided information about the impact of compounds **5** and **13** on the cell cycle of HT29 cells, showing their potential as cytostatic agents, although these results could also be related to the cell cycle arrest that accompanies apoptosis induction. Future studies will be needed to clarify this point.

#### 2.2.3. Induction of Apoptosis in Tumor Cell Lines

During the apoptosis process, the loss of the cytoplasmic membrane asymmetry occurs, caused by the translocation of phosphatidylserine (PS) from the leaflet of the internal membrane to the external membrane to be recognized by macrophage cells [23]. The exposed PS is recognized by annexin V phospholipid-binding protein, which binds to and fluorescently labels apoptotic cells.

To further investigate the potential mechanisms underlying the cytotoxic and cytostatic effects of compounds **5** and **13** on HT29 cells, Annexin V–FITC/PI double staining along with flow-activated cell sorter (FACS) cytometry analysis was employed for determinate apoptosis induction. This assay allowed for the distinction of different cell populations, including normal cells (Annexin V− PI−), early apoptotic cells (Annexin V+ PI−), late apoptotic cells (Annexin V+ PI+), and necrotic cells (Annexin V− PI+).

These assays on the HT29 cell line were conducted 72 h after treatment with the selected rearranged abietanes at their corresponding IC_50_ and IC_80_ concentrations. The results revealed significant apoptotic effects induced by both compounds on HT29 cells compared with the control group. Orthoquinone **13** induced apoptosis in 18% (6% early apoptosis and 12% late apoptosis) and 21% (7% early apoptosis and 14% late apoptosis) of the cell population at the IC_50_ and IC_80_ concentrations, respectively, as opposed to 9% of apoptotic cells in the control group (Figure 5).

Regarding pygmaeocin B (**5**), apoptosis induction was dose dependent, with 22% of total apoptosis at the IC_50_ concentration (5% early apoptosis and 17% late apoptosis), increasing to 35% at the IC_80_ concentration (12% early apoptosis and 23% late apoptosis). The percentage of necrotic cells was relatively high especially with treatment with compound **5** at the IC_50_ concentration compared with the control (15% vs. 3% for the control). Our results demonstrate the potent apoptotic effects of compounds **5** and **13** on HT29 cells. Further investigations are needed to elucidate the specific apoptotic trigger mechanisms involved in the apoptotic effects.

#### 2.2.4. Effects on Mitochondrial Membrane Potential (MMP)

Apoptotic effects induced by anticancer agents can be mediated through two major pathways: intrinsic and extrinsic apoptotic pathways. The intrinsic pathway involves the disruption of the mitochondrial membrane and changes in the mitochondrial membrane potential (MMP), while the extrinsic pathway leads to apoptosis without initial MMP alterations. To investigate the apoptotic responses of the HT29 cells treated with compounds **5** and **13**, the MMP changes were analyzed using flow cytometry staining with rhodamine 123 (Rh123) and IP after 72 h of treatment at IC_50_ and IC_80_ concentrations. The results showed that both orthoquinone **13** and pygmaeocin B (**5**) produced changes in the mitochondrial membrane potential with respect to the control (untreated cells) (Figure 6). Treatment with precursor **13** caused a decrease in positive Rh123-stained HT29 cells of around 40% at the IC_50_ concentration and 57% at the IC_80_ concentration, with the consequent increase in the negative cell population. Pygmaeocin B (**5**) caused a greater loss of membrane potential, with 54 and 61% decreases in Rh123+ cells with regard to the untreated control at IC_50_ and IC_80_ concentrations, respectively. This finding suggests that the apoptosis induction of compounds **5** and **13** in HT29 cells occurs through the activation of the intrinsic apoptotic pathway. Further investigations are necessary to elucidate the detailed molecular mechanisms underlying their apoptotic effect.

#### 2.2.5. Morphological Changes during Apoptosis Hoechst Staining

To assess the apoptosis induction capacity of compounds **5** and **13** on HT29 cells, we performed Hoechst 33242 nuclear staining, a well-established method for visualizing the morphological changes associated with apoptosis. Two principal morphological characteristics can be observed in the apoptosis process, chromatin condensation and nuclei fragmentation. Hoechst is a dye capable of penetrating cells and binding to double-stranded DNA, emitting blue light that can be visualized by using fluorescence microscopy.

After treating the colon adenocarcinoma cell line HT29 with diterpenoids **5** and **13** at IC_50_ and IC_80_ concentrations for 72 h, we examined the fluorescence images to evaluate the nuclear morphology (Figure 7). The control cells exhibited intact nuclear morphology, while a significant number of treated cells displayed apoptotic characteristics. At the IC_50_ concentration, both compounds caused the loss of normal nuclear architecture and chromatin condensation, as indicated by the higher concentration of Hoechst dye (bright blue). At the higher IC_80_ concentration, the chromatin condensation became irreversible, leading to the formation of pyknotic or contracted nuclei followed by fragmentation into apoptotic bodies (white arrow).

These results demonstrate that compounds **5** and **13** have remarkable apoptotic effects on HT29 cells inducing nuclear morphological changes characteristic of apoptosis. These findings support the potential use of these diterpenoids as promising agents for targeted apoptosis induction in colon adenocarcinoma cells. Further molecular studies will be necessary to confirm the underlying molecular mechanisms of the apoptotic activity of these products and to advance their potential applications in cancer therapy.

### 2.3. Anti-Inflammatory Activity

#### 2.3.1. RAW 264.7 Cell Viability

To evaluate the cytotoxic effects of the synthesized abietanes (**5**, **7**, **13**, **17**, and **19**) on RAW 264.7 monocyte/macrophage murine cells and to establish sub-cytotoxic concentrations for anti-inflammatory testing, we conducted cell viability assays using the MTT method. Each compound was tested at increasing concentration levels (0–100 μg/mL), and the IC_50_ values were determined by interpolation in the sigmoidal cytotoxicity curves (Figure 8). The obtained IC_50_ values for the different abietanes were as follows: 0.14 ± 0.09 μg/mL for pygmaeocin B (**5**), 2.81 ± 0.45 μg/mL for saprorthoquinone (**7**), 6.27 ± 0.94 μg/mL for pygmaeocin C (**17**), 9.29 ± 0.35 μg/mL for viridoquinone (**19**), and 8.18 ± 1.86 μg/mL for their precursor **13**. Additionally, sub-cytotoxic concentrations corresponding to ¾ IC_50_, ½ IC_50_, and ¼ IC_50_ were determined and selected for subsequent anti-inflammatory assays, ensuring that the observed effects were specifically attributed to the anti-inflammatory properties of the compounds rather than cytotoxicity.

These results demonstrate the cytotoxicity profile of synthesized abietanes on RAW 264.7 cells and provide the groundwork for investigating their potential anti-inflammatory effects at non-cytotoxic concentrations. The data support the selection of appropriate sub-cytotoxic concentrations for further anti-inflammatory experiments enabling a comprehensive evaluation of the compounds’ anti-inflammatory properties.

#### 2.3.2. Inhibition of Nitric Oxide (NO) Production

To evaluate the anti-inflammatory activity of the rearranged abietanes (**5**, **7**, **17**, and **19**) along with their precursor **13**, we conducted experiments on RAW 264.7 macrophage cells, a well-known model for screening anti-inflammatory compounds. The macrophages were stimulated with LPS to induce the maximum release of nitric oxide (NO) during the inflammatory response. Subsequently, the different compounds were incubated with the cells at sub-cytotoxic concentrations for 48 h (Figure 9). Next, we determined the concentration of nitrites in the culture medium, which was proportional to the amount of NO release.

The results demonstrated that all tested compounds exerted dose-dependent inhibition of NO release. At ¾ IC_50_ concentration, pygmaeocin B (**5**), saprorthoquinone (**7**), and viridoquinone (**19**) achieved 100% NO inhibition, indicating strong anti-inflammatory effects. Precursor **13** and pygmaeocin C (**17**) also significantly reduced NO release, with inhibition percentages of 99.6% and 91%, respectively. At ½ IC_50_ concentration, all compounds showed strong inhibitory activity, with inhibition percentages ranging from 94 to 96%. At ¼ IC_50_ concentration, the rearranged abietanes and their precursor displayed moderate to good inhibition of the inflammatory process, with inhibition percentages of NO release ranging from 42 to 64%.

Furthermore, to provide a comprehensive anti-inflammatory characterization of the compounds, we calculated the IC_50 NO_ values at 48 h of cell incubation (Figure 10). Among the compounds, pygmaeocin B (**5**) exhibited the highest effectiveness in inhibiting NO release in RAW macrophages, with an IC_50 NO_ of 33.0 ± 0.8 ng/mL, which was 61 times lower than that of its precursor **13** (2.03 ± 0.09 μg/mL). Saprorthoquinone (**7**) and pygmaeocin C (**17**) showed IC_50 NO_ values of 1.30 ± 0.08 and 1.73 ± 0.04 μg/mL, respectively. Finally, the IC_50 NO_ value for viridoquinone (**19**) was 7.21 ± 0.9 μg/mL.

These findings highlight the potent anti-inflammatory effects of synthetized abietanes (**5**, **7**, **17**, and **19**) and their precursor **13** on RAW 264.7 macrophage cells. The results support their potential as promising candidates for further development as anti-inflammatory agents, and their distinct mechanisms of action merit further investigation.

## 3. Discussion

Abietanes are a fascinating family of naturally occurring diterpenoids isolated from numerous plant species that have captured significant attention from the pharmacological community due to their diverse and intriguing biological potential. Over the past two decades, the scientific investigation of abietane and rearranged abietane-type diterpenoids has been a highly dynamic field of research. This extensive investigation has resulted in numerous publications focusing on the isolation, structural characterization, and preliminary biological studies of these compounds [24].

During our investigations, we found that the precursor, orthoquinone **13**, displayed the highest efficacy in inhibiting the proliferation of three tumor-tested cell lines. This remarkable result emphasizes the significance of modifying natural products to discover promising anticancer agents with superior bioactivity compared with their natural equivalents [25]. Structural analysis revealed that precursor **13** did not possess a rearranged abietane skeleton but presented orthoquinone functionality at C-11 and C12, which may have been responsible for its antiproliferative effects. These results align well with prior research on the cytotoxic activity of abietane-type diterpenes possessing a quinone moiety [26]; in this study, the researchers confirmed that the orthoquinone was the critical structural component for cytotoxicity in abietane diterpenoids with a naphthalene quinone moiety. They also demonstrated that tebesinone B and aegyptinone A (5.47–10.34 and 2.11–8.19 μM, respectively) as orthoquinones possessed higher activity than tebesinone A and aegyptinone B as para-quinones. Some other studies have also mentioned how drastically the cytotoxic activity of rearranged abietane diterpenes increases from the para to the ortho-orientation [27,28,29,30]. In addition, other research indicated the potent cytotoxic effect, against human carcinoma KB cells, of an orthoquinone-rearranged abietane, aethiopinone, whose structure is very close to that of saprorthoquinone (**7**) [31]. Significant cytotoxicity of aethiopinone has been demonstrated in other studies against human leukemia lymphoblastic NALM-6 (IC_50_ = 0.6 μg/mL) and promyelocytic HL-60 cells (IC_50_ = 4.8 μg/mL) [32].

Among the diterpenoids obtained in this study, pygmaeocin B (**5**), a rearranged abietane-type diterpenoid with an orthoquinone group on the C-ring, showed potent activity against human colon adenocarcinoma cells (HT29) and hepatocarcinoma tumor cells (HepG2). In contrast, viridoquinone did not display significant cytotoxicity in these cell lines, suggesting that the carbonyl group at the C2 position of pygmaeocin B plays a crucial role in enhancing the reactivity of the orthoquinone group (Figure 1). Additionally, another compound, saprorthoquinone (**7**), containing an orthoquinone moiety, exhibited significant cytotoxic effects against HT29 and murine melanoma cells (B16–F10). Other saprorthoquinone derivatives with biological properties have been described. In this context, a recent study showed the discovery of novel derivatives of saprorthoquinone (**7**) isolated from Salvia prionitis Hance, with dual inhibitor properties of Indoleamine 2,3-Dioxygenase 1 (IDO1) and Histone Deacetylase 1 (HDAC1). These results demonstrate that the orthoquinone moiety is a key pharmacophore for IDO1 and HDAC1 inhibition, providing a potential strategy for cancer treatment by exploiting both immunotherapeutic and epigenetic drugs [33]. Moreover, pygmaeocin C (**17**) with a catechol moiety that can be easily oxidized to form an orthoquinone showed decreased activity compared with pygmaeocin B (**5**). At the cellular level, orthoquinones can be enzymatically reduced by cytochrome P450, generating reactive oxygen species (ROS) capable of damaging the DNA and proteins of tumor cells [34], thus explaining the greater cytotoxicity observed in pygmaeocin B (**5**) (Figure 1).

The discovery and development of cytotoxic compounds with the ability to inhibit cell growth through cell cycle arrest and induction of apoptosis have been of great interest in the field of anti-cancer drug discovery [35,36,37,38,39]. In this study, two compounds, orthoquinone (**13**) and pygmaeocin B (**5**), were evaluated for their antiproliferative activity against the HT29 colon adenocarcinoma line. Both compounds showed significant antiproliferative effects, with pygmaeocin B (**5**) exhibiting greater potency. The mechanism of action of these compounds involves cell cycle arrest in the S phase and the induction of apoptosis, similar to other abietanes and related diterpenoids studied previously, such as tanshinone IIA [40] and triptolide [41].

Furthermore, apoptosis studies by AnnexinV-Fict staining together with mitochondrial membrane potential studies by Rhodamine 123 staining demonstrated that both compounds **5** and **13** induced apoptosis through the intrinsic apoptotic pathway at remarkably low concentrations, leading to a significant inhibition of HT29 cell viability. In addition, both compounds induced morphological changes characteristic of apoptosis, such as chromatin condensation and fragmentation, as evidenced by fluorescence microscopy. These findings agree with previous studies showing that some naturally occurring abietane diterpenes appear to induce apoptosis via the intrinsic pathway. For example, 6, 7-dehydroroyleanone [42] and rosmanol [43] were found to activate caspases 3 and 9, indicating their ability to induce apoptosis through the intrinsic cell death pathway. Therefore, further investigations are necessary to fully elucidate the mechanism responsible for the apoptotic effect of pygmaeocin B (**5**) and its precursor **13**.

Inflammation is a natural innate process of the immune system. However, when it persists for a prolonged period, it can trigger various chronic diseases including cancer [23]. The close relationship between colorectal cancer and the inflammatory state [44] prompted the analysis of the anti-inflammatory activity of the rearranged abietanes under study. According to the results, pygmaeocin B (**5**), with proapoptotic action against colon cancer cells, exhibited the highest anti-inflammatory activity. It inhibited NO production by the RAW 264.7 macrophage line at an extremely low IC_50 NO_ value. Orthoquinones **13**, **7**, and **19** and catechol quinone **17** followed pygmaeocin B (T) in terms of inhibition.

Previous studies have indicated that the inhibition of orthoquinone tanshinones abietanes diterpenes, including tanshinone IIA and cryptotanshinone, may decrease prostaglandin (PGE2) production via microsomal prostaglandin E2 synthase (mPGES-1), indicating a potential connection with their antiinflammatory and antiplatelet activities [45]. The immunomodulatory effects of abietanes paraquinone derivatives from the roots of Horminum pyrenaicum have been described; the extract fraction containing diterpene quinones horminones, specifically 15, 16-dehydroinuroyleanol, showed the most significant antiinflammatory effect on human peripheral blood mononuclear cells (PBMC) activated by phytohaemagglutinin [46].

Regarding the catechol group, nor-abietane diterpenoids extracted from Perovskia abrotanoides roots have shown great potential as anti-inflammatory agents, specifically catechol: deoxy-1,2-dien-3-oxoarucadio inhibited iNOS expression and NO production in LPS-stimulated J774A.1 macrophages [47].

Human cancer is a very complex disease, and therefore, more multifunctional drugs need to be developed. Efforts are needed to redirect research toward this goal. For example, the signaling pathway of the transcription factor NF-kβ is crucial to the interconnection between colorectal cancer and the inflammatory state, causing the activation of both proinflammatory and tumorigenesis-related genes. Furthermore, it highlights the role of the antiapoptotic members of the Blc-2 protein family [44]. NFkβ becomes a crucial therapeutic target to prevent and treat colorectal cancer. In future studies, it would be interesting to verify if pygmaeocin B (**5**) and other studied abietanes and rearranged abietanes exhibit a dual action through inhibition.

## 4. Materials and Methods

### 4.1. Chemistry

#### Experimental Procedures

The experimental procedure for the synthesis of compounds **5**, **13**, **14**, **15**, **16**, **17**, **18**, **19**, and **21** was previously described by our research group [21]. The experimental procedures for the synthesis of compounds **7**, **12**, and **20** and the ^1^H and ^13^C NMR spectra are included in the Appendix A (see Appendix A).

### 4.2. Biological Experimental Procedures

#### 4.2.1. Materials

Dulbecco2019s modified Eagle medium (DMEM), RPMI 1640 W/L-Glutamine, fetal bovine serum (FBS), penicillin/streptomycin (Biowest, Nuaillé, France), gentamicin (Biowest, Nuaillé, France), dimethylsulfoxide (DMSO, Merck Life Science S.L., Madrid, Spain), and 3-(4,5-dimethylthiazol-2-yl)-2,5-diphenyltetrazolium bromide (MTT) were purchased from Thermo Fisher Scientific Inc. (Ward Hill, MA, USA). Culture flasks and well plates were obtained from VWR International Ltd. (Radnor, PA, USA).

#### 4.2.2. Test Compounds

Firstly, the compounds (**5**, **7**, **13**, **17**, and **19**) were dissolved in DMSO at 5 mg/mL. Stock solution was stored at −20 °C before treatment and diluted in cell culture medium to appropriate concentrations for each experiment. For the antiproliferative assay in all cell lines, we determined the concentrations of compounds required for 20, 50, and 80% inhibition of cell growth, IC_20_, IC_50_, and IC_80_, concentrations, respectively, to analyze the full range of cytotoxicity and to determine the graded or acute response to these compounds. Subcytotoxic concentrations (¾ IC_50_, ½ IC_50_, and ¼ IC_50_), were used for the nitrite assay. All experiments were measured and compared with untreated control cells.

#### 4.2.3. Cell Culture and Viability Assay

The human colorectal adenocarcinoma cell line HT29 (ECACC no. 9172201; ATCC no. HTB-38), the human hepatocarcinoma cell line HepG2 (ECACC no. 85011430), the mouse melanoma cells B16–F10 (ATCC no. CRL-6475), and the murine monocyte/macrophage like RAW 264.7 cell line (ATCC no, TIB-71) were obtained from the cell bank of the University of Granada, Spain. The tumor cell lines were cultured in DMEM (Dulbecco’s modified Eagle’s medium) supplemented with 2 mM glutamine, 10% heat-inactivated FCS (fetal calf serum), and 50 μg/mL of gentamicin (for all cancer cell lines). The RAW 264.7 cell line was cultured in RPMI1640 medium, supplemented with 2 mM glutamine, 10% heat-inactivated FCS, and 50 μg/mL of gentamicin. All cell lines were incubated at 37 °C, in an atmosphere of 5% CO_2_ and 95% humidity. The culture media were changed every 48 h, and the confluent cultures were separated with a trypsin solution (0.25% EDTA). In all experiments, monolayer cells were grown to 80–90% confluence in sterile cell culture flasks.

The effects of the compounds on the cell viability were established using the MTT method [48] (Sigma, St. Louis, MO, USA). The cytotoxicity of the compounds was assessed by measuring the absorbance of MTT dye staining of metabolically competent cells. The cells were cultivated in 96-well plates at 6.0 × 10^3^ cells/mL for the HT29 and RAW 264.7 cell lines, at 5.0 × 10^3^ cells/mL for B16–F10 cells, and 15.0 × 10^3^ cells/mL for the HepG2 cell line, with a final volume of 200 μL per well being incubated with the different products (0–100 μg/mL). After 72 h, 100 μL of MTT solution (0.5 mg/mL) in 50% of PBS and 50% of the medium was added to each well. Following incubation for 1.5 h, formazan was resuspended in 100 μL of DMSO, and each concentration was tested in triplicate. The cell viability relative to the untreated control cells was determined by measuring the absorbance at 570 nm using an ELISA plate reader (Tecan Sunrise MR20–301, TECAN, Grödig, Austria). Compounds that had low IC_50_ values (5 and 13) were chosen for multiple cytometry assays, including the determination of apoptosis, cell cycle, and mitochondrial membrane potential determination.

#### 4.2.4. Measurement of Nitric Oxide Concentration

Nitrite production was analyzed using the Griess reaction method. The concentration of nitrite was used as an indication of nitric oxide (NO) production [49]. RAW 264.7 cells were seeded at 6 × 10^4^ cells/well in 24-well cell culture plates, supplemented with 10 μg/mL of lipopolysaccharide (LPS). After 24 h, the cells were treated with compounds (**5**, **7**, **13**, **17**, and **19**) at concentrations equivalent to ¾ IC_50_, ½ IC_50_, and ¼ IC_50_ of their half-maximal inhibitory concentration (IC_50_) for 48 h. The Griess reaction [50] was conducted by mixing 150 μL of supernatant test samples or sodium nitrite standard (0–120 μM) with 25 μL of Griess reagent A (0.1% n-(1-naphthyl) ethylenediamine dihydrochloride) and 25 μL of Griess reagent B (1% sulfanilamide in 5% of phosphoric acid) in a 96-well plate. The mixture was incubated at room temperature for 15 min, after which the absorbance was measured at 540 nm using an ELISA plate reader (Tecan Sunrise MR20–301, TECAN, Grödig, Austria). To determine the concentration of nitrite in each sample supernatant, the absorbance was referenced to the nitrite standard curve. To determine the percentage of NO production, the increase between the negative control (untreated cells) and the positive control (cells only with 10 μg/mL of LPS) was assigned 100%.

#### 4.2.5. HT29 Cell Cycle Analysis

The amount of DNA in the different phases of the cell cycle (G0/G1, S, and G2/M) was quantified using flow cytometry after staining with propidium iodide (PI) [51]. HT29 cells were seeded at a density of 5 × 10^4^ cells per well in 24-well plates containing 1.5 mL of culture medium. After 24 h the cells were treated with IC_50_ and IC_80_ concentrations of compounds **5** and **13** for 72 h. Next, the cells were washed twice with PBS, trypsinized, and resuspended in 1 × TBS (10 mM Tris and 150 mM NaCl). After this, Vindelov buffer (100 mM Tris, 100 mM NaCl, 10 mg/mL Rnase, 1 mg/mL PI, and 0.1% Triton x-100, pH 8) was added. Finally, the cells were kept on ice and stained with 20 μL of 1 mg/mL PI solution before being measured. Each experiment analyzed approximately 10 × 10^3^ cells. The experiments were performed twice, each with three replications per assay. The samples were analyzed using a flow cytometer, and the number of cells in each stage of the cell cycle was estimated by fluorescence-associated cell sorting (FACS) at 488 nm in an Epics XL flow cytometer (Coulter Corporation, Hialeah, FL, USA).

#### 4.2.6. Annexin V-FITC/Propidium Iodide Flow Cytometry Analysis

Flow cytometry was used to detect annexin V-FICT and PI double staining, to quantify the pro-apoptotic impact of compounds **5** and **13**. Apoptosis was assessed through flow cytometry using a FACScan flow cytometer (Coulter Corporation, Hialeah, FL, USA). In this assay, 5 × 10^4^ HT29 cells were plated in 24-well plates with 1.5 mL of medium and incubated for 24 h. The cells were then treated with the selected compounds in triplicate for 24, 48, and 72 h at their corresponding IC_50_ and IC_80_ concentrations. The cells were harvested and resuspended in a binding buffer (10 mM HEPES/NaOH with pH 7.4, 140 mM NaCl, and 2.5 mM CaCl_2_). Subsequently, Annexin V-FITC conjugate (1 µg/mL) was added and incubated for 15 min in the dark at room temperature. The cells were stained with 5 µL of 1 mg/mL PI solution before analysis. Approximately 10 × 10^3^ cells were analyzed in each experiment; the experiments were performed twice, each with three replications per assay.

#### 4.2.7. Flow Cytometry Analysis of the Mitochondrial Membrane Potential

We analyzed the electrochemical gradient across the mitochondrial membrane via analytical flow cytometry, using dihydrorhodamine (DHR). A total of 5 × 10^4^ HT29 cells were plated in 24-well plates for this assay. These cells were incubated for 24 h and then treated with compounds **5** and **13** at their corresponding IC_50_ and IC_80_ concentrations for 48 h. Next, the medium was replaced by adding fresh solution with DHR to a final concentration of 5 μg/mL. The cells were incubated at 37 °C for 1 h, washed, and resuspended in PBS with 5 μg/mL of PI. The fluorescence intensity was measured using a FACScan flow cytometer (fluorescence-activated cell sorter). The experiments were performed two times, each with three replications per assay.

#### 4.2.8. Hoechst-Stained Fluorescence Microscopy

The morphological changes were analyzed using Hoechst-stained fluorescence microscopy. For the analysis, 15 × 10^4^ HT29 cells were plated on a coverslip within a 24-well plate. After 24 h, compounds **5** and **13** were added to the cells and incubated at their respective IC_50_ and IC_80_ concentrations for the next 72 h. The cells were washed twice with PBS, treated with cold MeOH for 3 min, washed again with PBS, and then incubated in 500 μL of Hoechst solution (50 ng/mL) in PBS for 15 min in the dark. Finally, the samples were visualized through fluorescent microscopy (DMRB, Leica Microsystems, Wetzlar, Germany) under a DAPI filter.

#### 4.2.9. Statistical Analysis

The data were represented as the mean ± standard deviation (SD). For each experiment, the Student’s *t* test was used for statistical comparisons against the untreated control cells. A limit of *p* < 0.05 was used to determine the significant differences. *p* < 0.05 (*), *p* < 0.01 (**), and *p* < 0.001 (***). All data shown here were representative of at least two independent experiments, performed in triplicate.

## 5. Conclusions

In conclusion, the synthesis of some rearranged abietanes was carried out, including the synthesis of saprorthoquinone (**7**) and the first synthesis of 1-deoxyviroxocin (**12**) confirming its structure. Our biological study demonstrated the anti-inflammatory potential power of these compounds, especially the rearranged abietane pygmaeocin B (**5**) and its precursor orthoquinone **13**, Pygmaeocin C (**17**), and Viridoquinone (**19**), with very low IC_50NO_ values in (LPS)-induced RAW 264.7 cells. We also demonstrated the high anticancer potential of these compounds, especially of compounds **5** and **13**, in HT29 colon cancer cells with very low values of IC_50_ and with clear activation of apoptosis and intrinsic apoptotic pathway activation, as the results of apoptosis, cell cycle, and mitochondrial mem-brane potential have shown. The investigation of diterpenoids provided valuable insights into their cytotoxic and anti-cancer properties. These findings reveal the bioactive potential of the rearranged abietanes group for the first time, providing valuable knowledge to conduct further research of these natural products with respect to their anti-inflammatory and anticancer activities and, thus, study their detailed molecular mechanisms.

## Data Availability

Samples of compounds **7**, **12**, and **20** are available from the authors.

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
