# Peer review of "Evaluation of Anticancer and Anti-Inflammatory Activities of Some Synthetic Rearranged Abietanes"

_ijms, 2023, doi:10.3390/ijms241713583_

Round 1
Reviewer 1 Report
Mustapha Ait El Had et al. investigated the synthesis and the biological activity (anti-cancer and anti-inflammatory) of rearranged abietane-type diterpenes. The manuscript is interesting and has relevant elements for readers. However, some issues should be addressed before further consideration:
1. Lines 65-82: Improve the writing of this paragraph since it is challenging to be followed.
2. Line 84: Type instead of types.
3. Several passages have relevant typos. A detailed review throughout the manuscript looking for such typos is recommended.
4. Discussion should be improved since it is dedicated to describing results and including some introductory ideas, and a comparative discussion with previous studies and theories is missing.
5. Figure 6: This picture must be improved since the information is not adequately visualized.
6. In general, the quality and resolution of almost all figures must be improved.
7. Line 554: The origin, purity, and spectra of compound 13 must be adequately provided.
8. Conclusions summarize results, and conceptual findings from the mechanistic point of view must be provided for readers.
The manuscript has several grammar and stylistic issues to be revised since some passages are challenging to be followed. An editing language service is recommended to improve language issues.
Author Response
|
We thank the reviewer for their constructive comments, which helped us to improve the manuscript.
(1) Lines 65-82: Improve the writing of this paragraph since it is challenging to be followed. Response:
The paragraph has been well modified and improved. |
- Line 84: Type instead of types.
Response:
Corrected
|
(3). |
Several passages have relevant typos. A detailed review throughout the manuscript looking for such typos is recommended. |
Response:
According to the reviewer, the manuscript has been revised.
(4) Discussion should be improved since it is dedicated to describing results and including some introductory ideas, and a comparative discussion with previous studies and theories is missing.
Response:
The discussion has been improved according to the reviewer's suggestion.
In the chemical part, our synthesis of saprorthoquinone (7) has been compared with the synthesis described in the literature (line 117 and 121-122). in the biological part, this paragraph and three additional bibliographical references have been introduced.
In the biological part, this paragraph“ in this study, the researchers confirmed that the ortho-quinone is the critical structural component for cytotoxicity in abietane diterpenoids with a naphthalene quinone moiety. They also demonstrated that tebesinone B and aegyptinone A (5.47–10.34 and 2.11–8.19 μM, respectively) as ortho-quinones possess higher activity than tebesinone A and aegyptinone B as para-quinones. Some other studies have also mentioned how drastically the cytotoxic activity of rearranged abietane diterpenes increases from the para to the ortho-orientation [27-30]. In addition, other research indicated the potent cytotoxic effect, against human carcinoma KB cells, of an orthoquinone-rearranged abietane, aethiopinone ,whose structure is very close to saprorthoquinone (7) [31]” and three additional bibliographical references have been introduced in the text.
(5) Figure 6: This picture must be improved since the information is not adequately visualized.
Response:
According to the reviewer, the Figure 6 was improved
|
(6). |
In general, the quality and resolution of almost all figures must be improved. . |
Response:
According to the reviewer all figures have been improved
(7) Line 554: The origin, purity, and spectra of compound 13 must be adequately provided.
Response:
The synthesis procedure for orthoquinone 13 from ferruginol (1), along with the NMR data and its spectrum, has been added to the supporting information.
(8) Conclusions summarize results, and conceptual findings from the mechanistic point of view must be provided for readers.
Response:
According to the reviewer, the conclusion of the work was changed.
Reviewer 2 Report
This study presents the synthesis of rearranged abietane diterpenes and evaluates their anticancer and anti-inflammatory activities. Compounds 5 and 13 showed high cytotoxicity against HT29 cells, inducing apoptosis and cell cycle arrest. Pygmaeocin B (5) demonstrated potent anti-inflammatory effects. These findings suggest these compounds could offer a novel strategy for developing dual-function anticancer agents.
It is a good paper, however, there are areas for improvement in the paper.
Scheme 1: Synthesis of Pygmaeocins B (5) and C (17) from Ferruginol (1)
Scheme 1 depicts the stepwise synthesis process for obtaining pygmaeocins B (5) and C (17) from ferruginol (1). It is recommended that the authors include the structure of ferruginol (1) for comprehensive visualization.
Characterization Data:
The authors have solely presented NMR spectra for compounds at positions 8, 10, and 20. It is notable that this paper includes anti-cancer and anti-inflammatory evaluations for compounds 5, 7, 13, 17, and 19. However, there is a significant absence of structural characterization data such as NMR and Mass data throughout the study.
Compound Characterization:
While the procedure section provides NMR text for saprorthoquinone (7) and 1-deoxyviroxocine (12), it is recommended that NMR spectral data for these compounds be included. This additional inclusion will strengthen the characterization process.
The English writing in this paper was well done.
Author Response
We thank the reviewersfor their constructive comments, which helped us to improve the manuscript.
(1) Scheme 1 depicts the stepwise synthesis process for obtaining pygmaeocins B (5) and C (17) from ferruginol (1). It is recommended that the authors include the structure of ferruginol (1) for comprehensive visualization.
Response:
The structure of the product 1 has been added in the scheme 1
(2) The authors have solely presented NMR spectra for compounds at positions 8, 10, and 20. It is notable that this paper includes anti-cancer and anti-inflammatory evaluations for compounds 5, 7, 13, 17, and 19. However, there is a significant absence of structural characterization data such as NMR and Mass data throughout the study.
Response:
The spectra of products 5, 13, 17, and 19 have been included in the supplementary information. However, we have referenced the characterization of these products in the 'Experimental Procedures' section to the previous work conducted by our research group, as mentioned in reference [21].
(3) While the procedure section provides NMR text for saprorthoquinone (7) and 1-deoxyviroxocine (12), it is recommended that NMR spectral data for these compounds be included. This additional inclusion will strengthen the characterization process.
Response:
A numbering shift for certain products ( 8 10 ) has been corrected, while the NMR spectra of products 7 and 12 have already been presented in the supporting information
Round 2
Reviewer 2 Report
The resubmitted manuscript has been significantly improved, with all previously raised concerns appropriately addressed. The authors' dedicated efforts to enhance the paper are evident, resulting in a more coherent and polished work. Considering the substantial improvements made, I recommend accepting this manuscript for publication.